# A Drop-on-Demand Bioprinting Approach to Spatially Arrange Multiple Cell Types and Monitor Their Cell-Cell Interactions towards Vascularization Based on Endothelial Cells and Mesenchymal Stem Cells

**DOI:** 10.3390/cells12040646

**Published:** 2023-02-17

**Authors:** Joshua Weygant, Fritz Koch, Katrin Adam, Kevin Tröndle, Roland Zengerle, Günter Finkenzeller, Sabrina Kartmann, Peter Koltay, Stefan Zimmermann

**Affiliations:** 1Laboratory for MEMS Applications, IMTEK—Department of Microsystems Engineering, University of Freiburg, Georges-Koehler-Allee 103, D-79110 Freiburg, Germany; 2Department of Plastic and Hand Surgery, Medical Center—University of Freiburg, Faculty of Medicine, University of Freiburg, Hugstetterstraße 55, D-79106 Freiburg, Germany; 3Institute of Anatomy, University of Zurich, 8057 Zurich, Switzerland; 4Hahn-Schickard, Georges-Koehler-Allee 103, D-79110 Freiburg, Germany

**Keywords:** bioprinting, drop-on-demand printing, stem cells, endothelial cells, vascularization, biofabrication

## Abstract

Spheroids, organoids, or cell-laden droplets are often used as building blocks for bioprinting, but so far little is known about the spatio-temporal cellular interactions subsequent to printing. We used a drop-on-demand bioprinting approach to study the biological interactions of such building blocks in dimensions of micrometers. Highly-density droplets (approximately 700 cells in 10 nL) of multiple cell types were patterned in a 3D hydrogel matrix with a precision of up to 70 μm. The patterns were used to investigate interactions of endothelial cells (HUVECs) and adipose-derived mesenchymal stem cells (ASCs), which are related to vascularization. We demonstrated that a gap of 200 μm between HUVEC and ASC aggregates led to decreased sprouting of HUVECs towards ASCs and increased growth from ASCs towards HUVECs. For mixed aggregates containing both cell types, cellular interconnections of ASCs with lengths of up to approximately 800 µm and inhibition of HUVEC sprouting were observed. When ASCs were differentiated into smooth muscle cells (dASCs), separate HUVEC aggregates displayed decreased sprouting towards dASCs, whereas no cellular interconnections nor inhibition of HUVEC sprouting were detected for mixed dASCs/HUVEC aggregates. These findings demonstrate that our approach could be applied to investigate cell–cell interactions of different cell types in 3D co-cultures.

## 1. Introduction

Vascularization plays an important role in tissue-engineering applications aiming at the development of artificial tissue substitutes. In general, a microvascular network is essential to maintain healthy tissue. The network prevents cell death and allows the tissue to function correctly, through waste product removal and the supply of nutrients and oxygen [1]. Therefore, a vascular network is required for bioengineered constructs to ensure correct functioning of the tissue, to prevent cell death in constructs that are thicker than 200 μm, and to enable long-term viability [2]. Several bioprinting approaches aim to create a vascular network, for example via sacrificial printing, coaxial deposition or self-assembly of cells post-printing [2,3]. In this context, endothelial cells (ECs) such as human umbilical vein endothelial cells (HUVECs) and smooth muscle cells are often used together to fabricate blood-vessel-like structures [4,5,6,7]. Additionally, adipose-derived mesenchymal stem cells (ASCs/MSCs) are also used, because they are known to be able to secrete vascular endothelial growth factor (VEGF), an angiogenic growth factor, and it has been demonstrated that they can be differentiated into ECs and smooth muscle cells in vitro [8,9,10]. Since MSCs can be differentiated into several cell types, they are attractive candidates for the regeneration of several tissues. For example, they can be differentiated into osteogenic cell types, making them attractive candidates for bone-tissue engineering [11]. Additionally, HUVECs have been bioprinted with ASCs, and the printed scaffold led to blood-vessel formation as well as synthesis of a calcified bone matrix in vivo [12]. Considerable research has been devoted to better understanding the interactions between these cell types, but they are still not fully understood. For example, it has been demonstrated that ASCs support angiogenesis and promote lumen formation, whereas it has also been shown that co-culture of ASCs and HUVECs does not enhance angiogenic potential of HUVECs [13,14].

In 3D, spheroids are often used to investigate cell behavior since they mimic the natural cell environment more accurately than cells in 2D environments [15]. Using spheroids in hydrogels, active sprouting, migration and enhanced differentiation potential have been demonstrated [16,17,18]. The cells within spheroids interact with cells from neighboring spheroids through the secretion of growth factors or through physical contact, for example between sprouting/migrating cells [19]. Thus, the distance between spheroids is a crucial parameter for artificial tissues. However, there are few techniques that allow control of the spatial arrangement and the distance between two or more spheroids in a 3D environment. Controlling the distance between the spheroids allows better investigation of the cell–cell interaction between spheroids of different cell types, and various approaches have been developed. There is the kenzan method, which uses microneedles called “kenzans” to align spheroids, placement of spheroids based on aspiration, and more recent studies have reported micropatterning of hydrogels based on a stamp technique and subsequent placement of spheroids into the resulting gaps [18,20,21]. Previously, we developed a printing process to position HUVECs precisely in a fibrin hydrogel and showed that this approach can be used to compare high-density cell-laden droplets of HUVECs with HUVEC spheroids [3]. Furthermore, this study revealed that a droplet-based approach yielded longer cumulative sprout lengths (CSL) compared with spheroids, indicating a higher level of interaction between the cells. Therefore, the use of such droplets of high-density cell suspensions could reveal similar or even enhanced functional properties of spheroids, such as the sprouting behavior indicated above for endothelial cells, and also the self-assembly capacity of renal epithelial cells for the formation of tubular structures [22]. Therefore, a high-density cell-droplet approach could offer an alternative to the laborious spheroid manufacturing process and would enable the direct use of the cells after harvesting in a single-step biofabrication process. Based on this work, the aim of the study presented here was to expand this method, enabling printing of high-density cell-laden droplets of multiple cell types with high precision towards each other, in a 3D environment. This approach was used to investigate how different cell types influence each other. Droplets containing high densities of different cell types were printed onto a fibrin hydrogel and embedded into a sandwich-like structure to encapsulate the printed aggregates in a 3D environment. With the developed method it is possible to print different cell types in various constellations, ranging from a full overlap up to increasing distances between the cell. aggregates The patterns can be changed simply by adjusting the G-Code. The method was first established with immortalized mesenchymal stem cells (iMSCs) and was then expanded using additional cell types such as HUVECs, ASCs, and ASCs differentiated towards smooth muscle cells, to investigate the influence of the cell types on each other. We printed HUVECs and ASCs in various constellations via individual or mixed cell suspensions. Afterwards, the ASCs were substituted with ASCs differentiated into smooth muscle cells (dASCs) and the experimental setups were repeated to investigate the influence of the aggregates on each other, and to compare the behavior of HUVECs and ASCs.

## 2. Materials and Methods

### 2.1. Drop-on-Demand Bioprinter

A bioprinter equipped with a drop-on-demand (DoD) dispenser and a robotic stage that can move in x, y, and z directions was used for all experiments (Appendix A). It was equipped with a piezoelectric DoD dispenser (PipeJet nanodispenser, BioFluidix GmbH, Freiburg, Germany, more details described previously [23]) with exchangeable capillaries of 200 μm diameter. The dispenser was connected to a reservoir from which the bioink (between 50 and 150 μL) was filled by capillary forces. The dispenser was connected to a bioprinter with a three-axis robotic stage, which also incorporated an integrated optical system (SmartDrop, BioFuidix GmbH, Freiburg, Germany) to record and represent the shapes of the dispended droplets in flight. Based on the shape of the droplet, the volume could be calculated and the process parameters were adjusted accordingly to dispense a droplet with a volume of 10 nL. Due to batch-to-batch variations, the viscosity and surface tension of the bioink may have differed slightly between experiments, in turn influencing droplet generation. Thus, the optical system was also used to set parameters that allowed formation of a single droplet without any satellites, because satellites would decrease the accuracy of the printing process. The droplets were dispensed from the nozzle with a piezoelectric actuator-driven piston.

### 2.2. Cell Culture

All cell types were maintained at 37 °C with 5% CO_2_ in a humidified atmosphere using the specific cell culture medium. Passaging of cells was carried out when cells reached a confluency of less than 90%. After cells were harvested and samples were printed, all samples were incubated in endothelial cell growth medium (ECGM) (C-22010, PromoCell GmbH, Heidelberg, Germany) with 1% penicillin/streptomycin (P/S) (15140-122, Sigma-Aldrich, St. Louis, MO, USA) and 10% fetal calf serum (FCS) (10270-106, Sigma-Aldrich, St. Louis, MO, USA).

#### 2.2.1. HUVECs

Primary human umbilical vein endothelial cells (HUVECs) that were isolated from the vein of the umbilical cord of a single donor (C-12200, PromoCell GmbH, Heidelberg, Germany) were cultured in ECGM with 1% P/S and 10% FCS.

#### 2.2.2. Differentiated and Undifferentiated ASCs

Adipose tissue-derived stem cells (ASCs) were isolated from a 41-year-old female donor using an established protocol, as described previously [24]. Harvesting of ASCs was conducted with the informed consent of the patients, according to the Helsinki Declaration, and was approved by the institutional ethics committee. Cells were cultured in EBM^TM^-2 basal medium (CC-3156, LONZA, Basel, Switzerland) supplemented with EGM^TM^-2 SingleQuots^TM^ supplements (CC-4176, LONZA, Basel, Switzerland), 10% FCS and 1% P/S. ASCs from passages two to seven were used. ASCs were differentiated into smooth muscle cells (dASCs) according to a protocol that was reported previously [25]. In brief, differentiation was conducted as follows: At 90% confluency, ASCs were incubated in Minimum Essential Medium α (α-MEM) (32571028, Gibco^TM^ MEM α, Nukleoside, GlutaMAX^TM^ supplement, Thermofisher, Waltham, MA, USA) (with 1% P/S and 10% FCS). The media was supplemented with TGF-β1 (c_TGF_ = 5 ng/mL) over the course of seven days, and dASCs from passages three to seven were used.

#### 2.2.3. Immortalized Mesenchymal Stem Cells

Immortalized mesenchymal stem cells (iMSCs) were cultured in α-MEM with 1% P/S and 10% FCS. More information about iMSCs has been reported previously and can be found in the literature [26].

### 2.3. Bioink Preparation

To fabricate the bioink, cells were harvested and pelleted at 1000 rpm for 5 min. For HUVECs and iMSCs, fibrinogen (10 mg/mL, 341576, Sigma-Aldrich, St. Louis, MO, USA) was mixed with ECGM 1:1, yielding a final fibrinogen solution of 5 mg/mL in which the cells were suspended to a final concentration of 25 × 10^6^ cells/mL. For ASCs and dASCs, fibrinogen (10 mg/mL) was mixed with PBS 1:1 and the cells were suspended in the ink to obtain a concentration of 20 × 10^6^ cells/mL.

### 2.4. Hydrogel Matrix

Fibrin was chosen to embed the cell aggregates in a 3D environment. It was used as a substrate layer as well as a covering layer after printing, thus resembling a sandwich structure. It was prepared by using a pipette to mix 35 μL fibrinogen (10 mg/mL) and 35 μL thrombin (10 mg/mL, 605190, Sigma-Aldrich, St. Louis, MO, USA) in a 1:1 ratio. Fibrin was prepared on the 18 × 18 mm glass coverslips and samples were incubated at 37 °C in a humid atmosphere for 30 min before the printing process was started.

### 2.5. Process to Print Different Cell Types with High Precision on the Same Substrate

Bioink containing the first cell type was transferred into the dispenser. Then, the dispenser was left for 15 min during which the concentration increased to about 700 cells per 10 nL due to sedimentation (Appendix A) while the other bioink was kept on ice. During the 15 min, a droplet was ejected every three seconds to prevent clogging of the dispenser. After 15 min, glass slides with prepared fibrin hydrogel substrates were transferred onto a customized platform containing notches that exactly fitted the glass slides (Appendix A). The bioink was then printed onto the fibrin samples in the form of a predefined pattern that was set via G-code. When a second bioink with a second cell type was used, after printing the first cell type onto the samples, the samples were re-transferred into a humid atmosphere to prevent drying of the hydrogel, and the second cell type was filled into a new capillary. After leaving the second dispenser for 15 min with a droplet ejected every three seconds, the samples were again transferred onto the notches of the platform and the second bioink was printed in positions relative to the cell aggregates of cell-type one. Afterwards, the samples were incubated for 60 min during which the cell clusters adhered to the hydrogel due to enzymatic crosslinking of the fibrinogen in the bioink and the thrombin from the fibrin layer. During incubation over the following days, the cells developed structures that could be analyzed further, for example via microscope or antibody staining, to assess the influence of the different cell types on each other, e.g., enhanced growth towards a neighboring cell aggregate.

### 2.6. Printing Design and Quantification of Results

For this work, two print designs were used: printing aggregates of two cell suspensions next to each other or printing a line of several aggregates from the same cell suspension (Figure 1a). The relative positions of the cell types were described by either the pitch p (distance between the aggregates’ centers) or distance d (distance between the boundaries of the aggregates). An ROI was defined, which was a window of 60° towards the other cell type for cell pairs or two windows of 60° towards the neighboring cells for the line pattern. Thereafter, the images were edited with ImageJ and skeletonized (Figure 1d). The applied method yielded a projection of a 3D dimensional construct, so only statements relating to the imaged planes can be made. This was sufficient to determine the relative growth of cell types in the ROI and the other areas. At least two technical replicates with biological triplets as a minimum were created for each experiment. However, depending on the print design, the number of aggregates varied between three and twenty cell spots per technical replicate. Hence, the results are individually reported for each experiment. After skeletonizing, the CSL was measured in the ROI and the other region using ImageJ. As the ROI was smaller than the other region, the results from the other region were normalized to the ROI. Afterwards, a paired Student’s *t*-test was applied to determine statistical significance between the ROI and the other region. A probability value of * *p* < 0.05 indicated statistical significance, with increasing significances for ** *p* < 0.01 and *** *p* < 0.001.

### 2.7. Labeling and Imaging of Cells

CellTracker^TM^ was used to stain cell types differentially. CellTracker^TM^ Red CMTPX dye (ThermoFischer Scientific, C34552) was applied to label either ASCs or differentiated ASCs, while CellTracker^TM^ Green CMFDA dye (C2925, ThermoFisher Scientific, Waltham, WA, USA) was used to label HUVECs. Both dyes were also used to stain iMSCs. The dyes were dissolved in dimethyl sulfoxide (DMSO) (CAS 67-68-5, Sigma-Aldrich, St. Louis, MO, USA), and a stock solution with a final concentration of 10 mM was obtained and stored at −20 °C. The working solution was prepared by diluting the stock solution to a final concentration of 5 μM in cell-specific serum-free cell culture medium. After cells were harvested and pelleted, the pellet was suspended in 2 mL of the working solution and incubated for 40 min at 37 °C. Afterwards, the cells were washed and centrifuged, the supernatant was removed and the cells were suspended into the bioink. The fluorescent images were captured with an Observer Z1 microscope (Zeiss, Germany, Jena, Germany) and processed using the software Zen blue edition (Zeiss, Jena, Germany). Excitation wavelengths of 470 nm and 555 nm for CellTracker^TM^ Green and CellTracker^TM^ Red staining, respectively, were used for image acquisition.

### 2.8. Live-Dead Assay

To assess the influence of the printing process on the viability of the cells, a live–dead assay was performed immediately after encapsulating the cells with the second fibrin layer. Samples were stained with a PBS-based staining solution of 2 μM calcein-AM from Thermo Fisher Scientific (Waltham, MA, USA) for viable cells and 6 μM ethidium homodimer-1 from Sigma Aldrich (St.Louis, MO, USA) for dead cells, for 30 min at room temperature in the dark. Cells pipetted manually and not printed via DoD were used as the positive control. For the negative control, cells were immersed in 80% ethanol for 30 min. Fluorescence images were obtained with an Observer Z1 microscope with excitation wavelengths of 470 nm for calcein-AM and 555 nm for ethidium homodimer-1, using the software Zen blue edition. To ensure a consistent evaluation, an approach as previously published was used to assess the signals [27]. The signals were transformed into pseudocolors, and only signals with intensity values higher than 70% of the maximum intensity were counted. Finally, the cell viability was calculated by dividing the number of viable cells by the number of total counted cells.

### 2.9. Immunostaining

Cells were fixed with ice-cold methanol and incubated for 30 min with a blocking solution (3% BSA; 0,1% Tween 20 in PBS), followed by incubation with the primary antibody, monoclonal mouse anti-alphaSMA antibody (Dako Agilent, Santa Clara, CA, USA) diluted 1:50 in blocking solution for 24 h at 4 °C. After washing with PBS, the corresponding secondary antibody (goat anti-mouse IgG conjugated with Alexa Fluor 568, Thermo Fisher Scientific, Waltham, MA, USA) diluted 1:200 in PBS was applied, followed by incubation for another 24 h at 4 °C. Thereafter, cell nuclei were stained with DAPI (DAPI 33342, Hoechst Thermo Fisher Scientific, Waltham, MA, USA) for 20 min at RT. Cells were then washed three times with H_2_O, and fluorescence images were obtained with an Observer Z1 microscope and the software Zen blue edition.

## 3. Results and Discussion

### 3.1. Process to Print Different Cell Types with High Precision on the Same Substrate

The developed process is shown schematically in Figure 2. In this process, the first cell type is printed onto the fibrin substrate with a pattern that is pre-defined via G-Code. Afterwards, a second cell type is printed onto the fibrin substrate containing the first cell type, and printing patterns are designed in relation to the first cell type. Then, a second layer of fibrin is added on top, and cells are incubated in ECGM cell culture medium for several days. The samples can be then analyzed, for example via antibody staining or visual analysis. In one experiment, between N = 3–15 samples with n = 6–48 cell aggregates were fabricated via the described process, demonstrating the capability to pattern numerous hydrogels with a high number of cell aggregates. A droplet-based technique displays the advantage of printing aggregates with defined patterns, including full and partial overlaps, requiring little effort in preparation. In contrast with approaches using spheroids, such as aspiration-based techniques [21] or the kenzan method [20], the cells were directly harvested from the culturing flasks, suspended into the bioink and printed. Therefore, the presented approach does not require the labor-intensive and time-consuming process that is necessary to fabricate spheroids. In addition, the process allows fully automated patterning of the substrates via G-Code after the bioink is transferred into the dispenser. Furthermore, in contrast with other methods, such as kenzan [20] or stamp-based approaches [18], the gap between the cell aggregates can easily be varied by adjusting the G-Code. This allows the use of flexible adjustments within a single experiment. For this work, HUVECs, ASCs and ASCs differentiated into SMCs were used as they are relevant for the engineering of vascularized tissue. However, the established method could also be applied to other cell types, which are relevant for different tissues, including stromal cells like fibroblasts for additional structural functions.

### 3.2. Printing Performance

Several experiments were performed to assess the printing performance in terms of robustness, precision, and cytocompatibility (impact on cell viability). First, patterns containing arrays of 6 × 8 iMSC cell aggregates with pitches (distance from center to center of the cell aggregates) of 500 μm between them were fabricated on N = 3 samples (Appendix A). The aggregates had a circular shape and a radius of r = 125 ± 5 μm before covering. After adding the second fibrin layer, a radius of r = 126 ± 6 μm was measured. An overlay of images from before and after covering showed no displacement or disintegration of the cell spots. Thus, it was concluded that adding a second layer does not influence shape fidelity of cell aggregates.

Second, a live–dead assay was performed with iMSCs post-printing to characterize the influence of the dispensing technique on cell viability by varying the stroke of the piezo-actuator or the stroke velocity of the piezo-actuator. For a constant stroke of 21.6 µm, the stroke velocity increased from 30 μm/ms to 120 μm/ms in 10 μm/ms steps, and for a constant stroke velocity of 60 μm/ms, the stroke was increased from 10.8 µm to 36 µm in 3.6 µm steps. For each measurement point n = 10 cell spots were analyzed. We analyzed this range of parameters because batch-to-batch variations in surface tension and viscosity led to slightly different printing parameters for the generation of a 10 nL droplet without any satellites. The printing process was shown to have only a minor negative effect on cell survival. For example, the parameters selected to generate droplets with a volume of 10 nL reduced cell viability to only about 80%, compared to the non-printed positive control of 85%. Results are shown in Appendix A. Cell behavior and reorganization post-printing was observed for up to 4 days in later experiments, because it was previously revealed that such a time frame is sufficient to investigate the reorganization of high-density cell-aggregates after printing [3]. The influence of an analogous drop-on-demand bioprinting approach on the viability of HUVECs has previously been investigated [3]. To determine the precision of the developed process, iMSCs were stained with Celltracker^TM^ (C34552, Thermo Fisher Scientific, Waltham, MA, USA) green and red, representative for two different cell types in future experiments. A pattern was printed (as shown in Figure 3a), in which red iMSCs were printed in a line with a pitch of $p_{x,\;r}=1000\;\mu m$ and $p_{r,\;y}=0\;\mu m$ between the aggregates. Afterwards, green iMSCs were printed in positions relative to the red iMSCs, where the green iMSCs had an ideal pitch of $p_{x,\;g}=0\;\mu m$ and $p_{y,\;g}=800\;\mu m$ from the respective red iMSC cell aggregate (indicated by the yellow circles in Figure 3a). N = 8 technical replicates with n = 21–32 cell-pair aggregates were fabricated. Afterwards, the deviations Δx and Δy of each cell aggregate from their ideal positions were analyzed. Results for the deviations of the red iMSCs from their ideal positions are shown in Figure 3b, where the symbol represents individual spots within a run and colors represent different runs. The average deviations in the x and y directions were obtained as $\bar{{\Delta x}_{r}}=0\pm 29\;\mu m$ and $\bar{{\Delta y}_{r}}=3\pm 25\;\mu m$.

The locations of the green cell spots were analyzed in two separate experiments. First, the glass slides with the hydrogels were not removed from the printing platform between printing the red and green iMSCs. This yielded average deviations from the ideal position of $\bar{{\Delta x}_{g}}=78\pm 69\;\mu m$ $\bar{{\Delta y}_{g}}=1\pm 60\;\mu m$. The results are shown in Figure 3c. In the second experiment, the hydrogels were removed from the printing platform after printing the red iMSCs, then transferred into a humid atmosphere and re-transferred onto the printing platform when the green iMSCs were printed. This process increased the deviations from the ideal position $\bar{{\Delta x}_{g}}=-214\pm 171\;\mu m$ and $\bar{\Delta y_{g}}=223\pm 92\;\mu m$. Results are shown in Figure 3d. The increase in average deviation was probably caused by the transfer process of the samples. However, within one run the average alignment errors on the individual samples were ±58 μm in the x direction and ±54 μm in the y direction. This is also indicated by the clustering in the results shown in Figure 3c,d. By monitoring the printing process after each print it was possible to print defined intervals between the cell types regardless of the initial alignment errors, for example from a full overlap up to a pitch of 1000 μm (Figure 3e).

Based on these results, a printing process without transferring the samples between the printing of different cell types would yield lower errors. However, in order to prevent evaporation of the fibrin hydrogel, samples were temporarily stored in the humidified incubator between printing the two cell types. This issue could be addressed in further research, e.g., by implementing evaporation protection in the bioprinting device.

Altogether, we were able to demonstrate the possibility of aligning different cell types with pre-defined distances by simply adjusting the G-Code. Especially, Figure 3e shows that it is possible to print high-density cell-laden droplets with a partial overlap (220 μm group). To our knowledge, this has not previously been achieved with high-density cell-laden droplets and is not feasible for methods based on spheroids, which are already self-contained entities. Such partial cell contact may allow direct comparison of the behavior of cells in contact with a second cell type and cells in the same aggregate without contact with a different cell type. This is of particular interest in the investigation of developmental biological processes, such as vascularization, in which cells are constantly interacting with their microenvironment. The following sections describe results based on relevant cell types involved in such processes.

### 3.3. Influence of ASCs and HUVECs onto Each Other

After establishing and characterizing the process, red-labeled ASCs and green-labeled HUVECs were printed. They were either suspended in two separate cell suspensions or in the same bioink as a mixed cell suspension.

#### 3.3.1. ASCs and HUVECs as Separate Cell-Laden Droplets

N = 2 technical replicates with a total of $n_{\mathrm{full}\;\mathrm{overlap}}=11$ pairs of cell aggregates were printed and imaged daily for four days (Figure 4a). The distance between each pair of cell aggregates was 4000 μm, to avoid their influencing each other. On day two, HUVECs started to develop sprouts, whereas ASCs developed structures that appeared similar to sprouts and were thus termed sprout-like structures. For each cell type, these structures grew larger in the subsequent two days. It was observed that ASCs did not inhibit the sprout formation of HUVECs. This stands in contrast to recent findings, where the presence of spheroids containing human mesenchymal stem cells inhibited sprouting from HUVEC spheroids [21]. These differences may be explained by the different sources of the stem cells and the fact that in the previously mentioned study the spheroids were not in direct contact with each other. Here, on the contrary, printing with a full overlap led to joint sprout-like structures in each sample, which appeared to be formed by HUVECs and ASCs together (indicated by the white arrows in Figure 4b). Such joint structures were detected on all pairs of cell aggregates. It may be possible that HUVECs recruited ASCs, since it has been shown that ASCs can support the formation of vascular-like structures in vitro [28]. Nonetheless, most structures appeared to consist entirely of either HUVECs or ASCs instead of joint structures. Future studies could further investigate this phenomenon and the spatial arrangement of the cell types.

Next, ASCs and HUVECs were printed with a partial overlap (Figure 4c) and imaged for four days. N = 2 constructs with a total of $n_{\mathrm{partial}\;\mathrm{overlap}}=11$ cell pairs were fabricated. On day two, the HUVEC aggregates formed clear sprout structures that appeared to permeate the ASC aggregates, as indicated by the white arrows in Figure 4c, day two. Interestingly, most of the HUVEC aggregates showed clear sprouts only in regions without ASCs one day later, at day three (indicated by white arrows in Figure 4c, day three), which suggests that the direct cell–cell contact with ASCs may enhance sprout formation towards ASCs, or that fibrin remodeling is more manageable around the ASC cell aggregate. ASCs also formed sprout-like structures that appeared to form preferably towards the HUVECs, as again indicated by the white arrows. On day four, ASCs could be found inside the HUVEC spots, where no signal had been detected on day zero.

Finally, ASCs and HUVECs were printed in proximity in groups with three different distances (distance between boundaries of the cell spots: 203±24 μm, 372±47 μm and 527±57 μm).

For each distance, N=2 technical replicates were printed with total amounts of $n_{200\;\mu m,\mathrm{ASCs}}=6$,$\;n_{200\;\mu m,\mathrm{HUVECs}}=6$, $n_{370\;\mu m,\mathrm{ASCs}}=5$, $n_{370\;\mu m,\mathrm{HUVECs}\;}=6$, $n_{530\mu m,\mathrm{ASCs}\;}=8$, $n_{530\;\mu m,\mathrm{HUVECs}}=8$ cell aggregates. Representative images for each group are shown in Figure 5a–c. As described in detail in the methodology, a region of interest (ROI) of 60° was defined towards the neighboring cell aggregates, and the CSL was measured in the ROI and the other region (300°). Afterwards, the results were normalized to 60° to achieve better comparability. Measurements were performed on day three post-printing and results are shown in Figure 5d.

In the 200 μm group, there was significantly less sprouting from HUVECs towards the ASCs, whereas there were significantly more sprout-like formations from the ASCs towards the HUVECs. This behavior decreased with increasing distances between the cell types and no significant differences were observed in the other groups.

The increased sprout-like formation of ASCs towards HUVECs was consistent with the results from the partial overlap, where ASCs grew and formed structures increasingly oriented towards HUVECs. Released stimulation proteins (e.g., growth factors) from HUVECs may recruit ASCs, which may be one reason why the increased growth from ASCs towards HUVECs weakened with increasing distances [29]. Interestingly, while the behavior of ASCs was consistent for a gap between the cell types and a partial overlap, the behavior of HUVECs appeared to change as soon as cell contact between ASCs and HUVECs was no longer present. While there was clear sprout formation from HUVECs when printed overlapping with ASCs, for close distances the opposite behavior was observed. These differences suggest that direct cell–cell contact may yield different behaviors of cells in the aggregates. It is known that direct contact between the cells can change their behavior. For example, it has been reported that direct cell–cell contact between ASCs and HUVECs leads to increased expression of αSMA or release of activin A [29,30]. In addition, printing two cell types onto the same spot may lead to local differences in cell–hydrogel interactions which in turn may change cell behavior. Since the exact reasons for this behavior could not be identified, further research is required to understand the underlying causes. However, these findings should also be taken into account when designing larger tissue structures containing the applied cell types.

#### 3.3.2. ASCs and HUVECs as Mixed Cell Suspension

After analyzing the behavior of ASCs and HUVECs printed as separate aggregates, their behavior was investigated when mixed into the same bioink and dispensed as mixed cell-laden droplets. Analogously to a recent study, they were mixed at a ratio of 2 (HUVECs) to 1 (ASCs) [31]. Mixed cell-laden droplets were printed with pitches of 500 μm, 800 μm, 1000 μm and 2000 μm and were imaged every day for three days. Representative images of the cell aggregates for each distance and day are shown in Figure 6a–d. Per pitch, N = 2 technical replicates were printed, except for the 1000 μm group with N = 3. In total, the number of corridors between cell spots were $n_{500\;\mu m}=42$, $n_{800\;\mu m}=26$, $n_{1000\;\mu m}=32$ and $n_{2000\;\mu m}=9$. As a positive control for the behavior of HUVECs, HUVECs without ASCs were printed with the same pitches. In total, $N_{\mathrm{HUVECs}}=8$ technical replicates with $n_{\mathrm{HUVECs}}=54$ aggregates were printed. Similarly, as a positive control for ASCs, ASCs without HUVECs were printed with the same pitches (Appendix A). During the three days, the ASCs mainly developed sprout-like structures. In the 500, 800 and 1000 μm groups, clear directionality towards neighboring cell aggregates was visible. In the 500 μm group, connections were formed with the neighboring cell aggregates after only one day. This trend was also clearly visible in the 800 μm and 1000 μm groups, where direct connections formed on days two and three. A representative image of the 800 μm group on day three is shown in Figure 6e. The influence of the pitch was further demonstrated by analyzing the CSL. As shown in Figure 6f, the proportion of structures in the ROI decreased with increasing distances. For the 500 μm pitch, the CSL on day one was more than 700% higher in the ROI than in the other region. The proportions on day one decreased to about 300% in the 800 μm group and about 60% in the 1000 μm group. Over the next two days, these proportions slightly increased (Appendix A). In addition, in the 2000 μm group the CSL in the ROI was significantly higher than in the other region on days two and three, whereas no significance was observed on day one.

The structures formed by ASCs may be cellular bridges. Such bridges can form between ASC and ASC+HUVEC spheroids, as was recently reported [18]. These results also further support our previous findings that high-density cell-laden droplets may behave somewhat like spheroids. The same experimental setup containing ASCs without HUVECs also showed directional growth of ASCs, although no such bridges and no relation to the pitch were observed (Appendix A). The first finding stands in contrast to reports that pure ASC spheroids also formed such bridges [18].

After the ASCs started to form sprout-like structures, HUVECs also occasionally developed sprouts. A representative image is shown in Figure 6g. It is clearly visible that HUVEC sprouts aligned and organized themselves depending on the ASC structures. While 98% of HUVEC aggregates in the positive control (pos. cont.) (Figure 6h) developed homogenous sprouts with no preferred sprouting direction, only 45% of HUVECs in mixed cell-laden aggregates developed sprouts. This difference in behavior may be caused by inhibitory effects of the stem cells on HUVECs [32]. However, these observations stand in contrast to the behavior of HUVECs and ASCs printed as separate droplets on top of each other, as discussed above. These different behaviors may be caused by different interactions between the cells caused by different cell distributions when mixing or printing, but further research is required. As can be seen, ASCs formed more sprout-like structures than HUVECs, which is interesting since there were approximately twice as many HUVECs as ASCs in the bioink. However, since the applied method does not allow analysis of the three-dimensional spatial arrangement of the cells, exact statements cannot be made. Furthermore, the reason for the formation of cellular bridges is not yet clear and further research is required.

### 3.4. Influence of Smooth Muscle Cell Differentiated ASCs and HUVECs on Each Other

After the behavior of ASCs and HUVECs was analyzed, ASCs were substituted with ASCs that were differentiated into a different cell type, namely smooth muscle cells (dASCs). This cell type was chosen due to its frequent use with endothelial cells for fabrication of vascularized structures, as discussed above. The same experimental set-ups as with HUVECs and ASCs were chosen to evaluate similarities and differences between the behaviors of the different cell types.

#### 3.4.1. Differentiation of ASCs

ASCs were differentiated into smooth muscle cells by TGF-beta1 treatment for 1 week. In comparison to ASCs without TGF-beta1 treatment (Figure 7a), TGF-beta1 induced the expression of alpha smooth muscle actin (αSMA), which is indicative of the differentiation of ASCs towards smooth muscle cells (Figure 7b).

#### 3.4.2. dASCs and HUVECs as Separate Cell-Laden Droplets

Differentiated ASCs (dASCs) and HUVECs were printed in similar experimental setups to the undifferentiated ASCs and HUVECs (see Section 2.2.1). For a full overlap, N = 2 technical replicates with a total of $n_{full\;overlap}=8$ cell pairs per distance were printed. A representative image from day three is shown in Figure 7c. As with ASCs and HUVECs, over the course of three days both cell types developed sprouts or sprout-like structures with no preferred direction for either cell type. For a partial overlap, N = 2 technical replicates with a total of $n_{partial\;overlap}=8$ cell pairs were printed. A representative image is shown in Figure 7d. While both cell types again developed sprout(-like) structures, no growth of HUVECs towards the dASCs was observed in this constellation, which stands in contrast to findings from printed ASCs and HUVECs. This may be explained by different interactions of HUVECs in contact with dASCs and ASCs. Finally, the two cell types were printed with increasing distances and no contact in three different groups (distances between boundaries of the cell aggregates: 208±26 μm, 435±33 μm and 605±30 μm, Figure 7e–g). N = 2 technical replicates with a total of n_separate = 8–15 cell pairs were printed and the CSLs were analyzed on day three. For HUVECs in the 210 and 435 μm group, significantly fewer sprouts formed in the ROI towards the dASCs. This behavior is consistent with observations of HUVECs and ASCs, as discussed above. In contrast to ASCs, HUVECs did not appear to influence the directionality of the sprout-like structures developed from the dASCs and no trend was visible. This was surprising, as endothelial cells are known to recruit smooth muscle cells [33]. Reasons for the anti-directional formation of HUVEC sprouts are not yet known and require further research. However, as smooth muscle cells are often used with HUVECs for biofabrication of vascularized structures, this behavior can already be considered for possible bioprinting designs.

#### 3.4.3. dASCs and HUVECs as Mixed Cell Suspension

dASCs and HUVECS were mixed (ratio 1:2) and printed with pitches of 500 μm, 800 μm, 1000 μm and 2000 μm. In total, N = 2 technical replicates (except the 2000 μm group with N = 1) with n = 6–8 cell pairs per replicate were fabricated. Images captured on day three are shown in Figure 8a–d. As previously, joint structures of both cell types were formed, as indicated by the white arrows in Figure 8e. In contrast to ASCs and HUVECs, no direct connections between the cell spots were visible, which may have been caused by differentiation of ASCs into a different cell type. Also, all HUVECs displayed clear sprouting behavior, not merely a few single sprouts as was the case for the mixed cell suspension of HUVECs and ASCs. Thus, both cell types were analyzed for directed sprout formation towards the neighboring cell clusters and no directionality was observed (Figure 8f). These observations suggest that in mixed cell suspensions each respective cell type has no inhibitory impact on the other cell type and that their post-printing behavior does not necessarily depend on neighboring cell aggregates. These observations are contrary to the behavior observed in the experiments conducted with ASCs and HUVECs, discussed above, and the differences may result from the differentiation of ASCs into smooth muscle cells, in turn leading to different interactions with endothelial cells. Therefore, our findings indicate that bioprinting of pre-differentiated ASCs is beneficial for the creation of vascularized tissues. In the future, further methodologies such as confocal microscopy, immunostaining and gene expression analysis could be applied to analyze the spatial arrangement of the individual cell types.

## 4. Summary and Conclusions

To fully yield the potential of future bioprinting approaches, there is an increasing necessity to understand how cells influence each other in 3D environments post-printing, and which spatial patterns should be created by bioprinting. However, few approaches allow precise positioning of high-density and locally confined cell aggregates in 3D. Here, an approach based on a sandwich technique was used, which allowed high-precision deposition of high-density cell-laden droplets from two different cell types onto a fibrin hydrogel. Average alignment errors between runs were approximately 250 μm, and errors within runs were less than 60 μm. ASCs were printed with HUVECs as individual cell aggregates and significant less sprouting from HUVECs towards ASCs was observed with a gap of 200 μm between them, whereas at that distance ASCs formed significantly more structures oriented towards HUVECs. Partially reduced sprouting from HUVECs towards ASCs is particularly interesting because it has been reported that the presence of an MSC (of which ASCs are a subgroup) spheroid in close proximity to a HUVEC spheroid leads to complete inhibition of HUVEC sprouts [21]. Directed growth of ASC structures towards HUVECs may be explained by recruitment of ASCs by HUVECs, which has been demonstrated previously [28]. In mixed cell suspensions, ASCs formed directed structures and cellular bridges between the aggregates at pitches up to 1000 μm, and potential inhibitory effects of ASCs on HUVECs were observed. Such bridges were also recently reported for ASC+HUVEC spheroids with inter-spheroid distances up of 200 μm [18]. These differences relating to distances and the mechanisms leading to bridge formation require further investigation. The same experiments were repeated with ASCs that were differentiated into smooth muscle cells. For these cell types, significantly fewer sprouts were observed from HUVEC aggregates towards dASCs, whereas the HUVECs appeared to have no measurable impact on the dASCs. For mixed cell suspensions, no directionality was observed for any cell type and all HUVECs displayed good sprouting behavior. For ASCs as well as dASCs in combination with HUVECs, joint structures containing two cell types were often observed.

To conclude, these findings demonstrate the capabilities of the developed method to discover new phenomena relating to the interactions between different cell types. In addition, our approach addresses several shortcomings of other methods for spatial alignment of high-density cell aggregates. For example, a droplet-based technique does not require the labor-intensive process necessary to fabricate spheroids, and the printed pattern can be adjusted as required by simply changing the pattern via G-Code. Such flexible adjustments are not possible with the kenzan method or stamp-based techniques because of the pre-defined distances of the kenzans or notches. Aspiration-based approaches require manual picking and placing of individual spheroids, making it challenging to automize the printing process. In addition, it is not feasible to print a partial or full overlap using spheroids, whereas this is possible with the presented method. While the reported experiments were conducted with cell types relevant for vascularization, the approach can be easily transferred to other cell types to investigate the interaction between cells that are relevant for other tissues.

In the future, the developed approach should be combined with other methodologies to explain the causes of the observed cell behaviors, which are not yet fully understood. Nonetheless, directional and anti-directional growth of cell types can already be considered when biofabricating tissues and structures using the investigated cell types, and the method could be also applied to investigate the interactions between other cell types.

## Figures and Tables

**Figure 1 cells-12-00646-f001:**
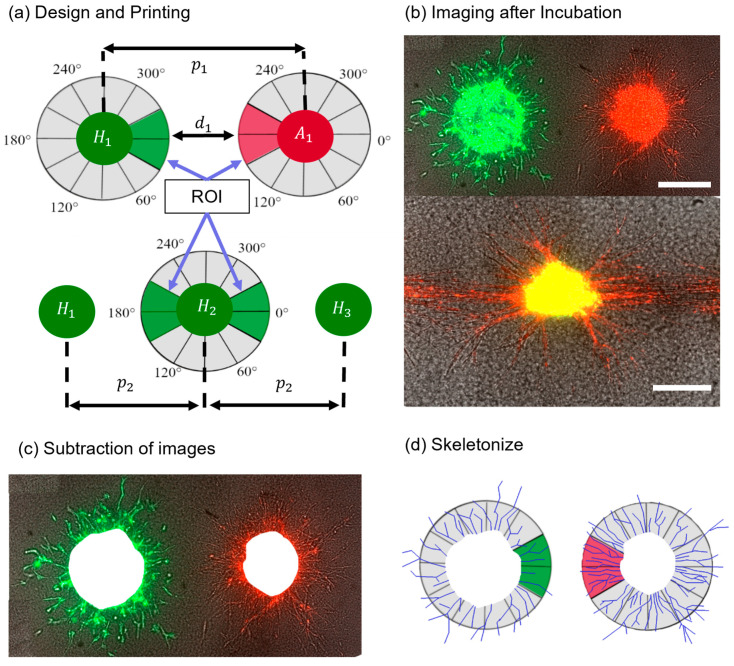
Schematic of the steps from printing design to analysis. First, (**a**) one of the print designs was fabricated, then (**b**) aggregates were imaged after several days of incubation. Next, (**c**) the images from day 0 were subtracted and the aggregates were skeletonized (**d**) to analyze the structures in the ROI and other regions. Scale bars: 200 μm.

**Figure 2 cells-12-00646-f002:**
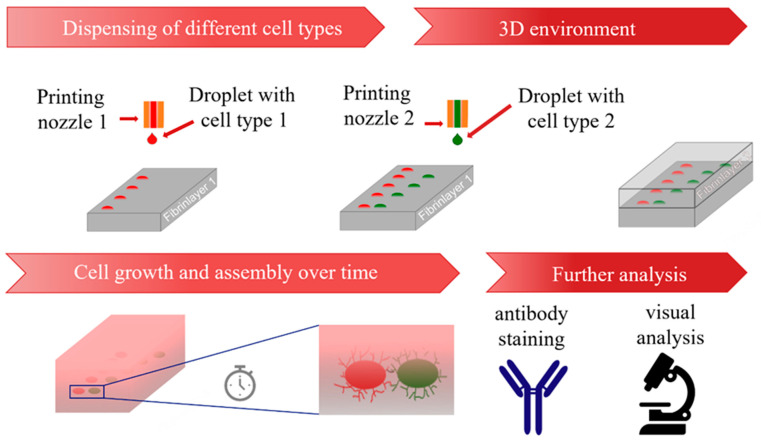
Schematic representation of the developed process from printing process to analysis.

**Figure 3 cells-12-00646-f003:**
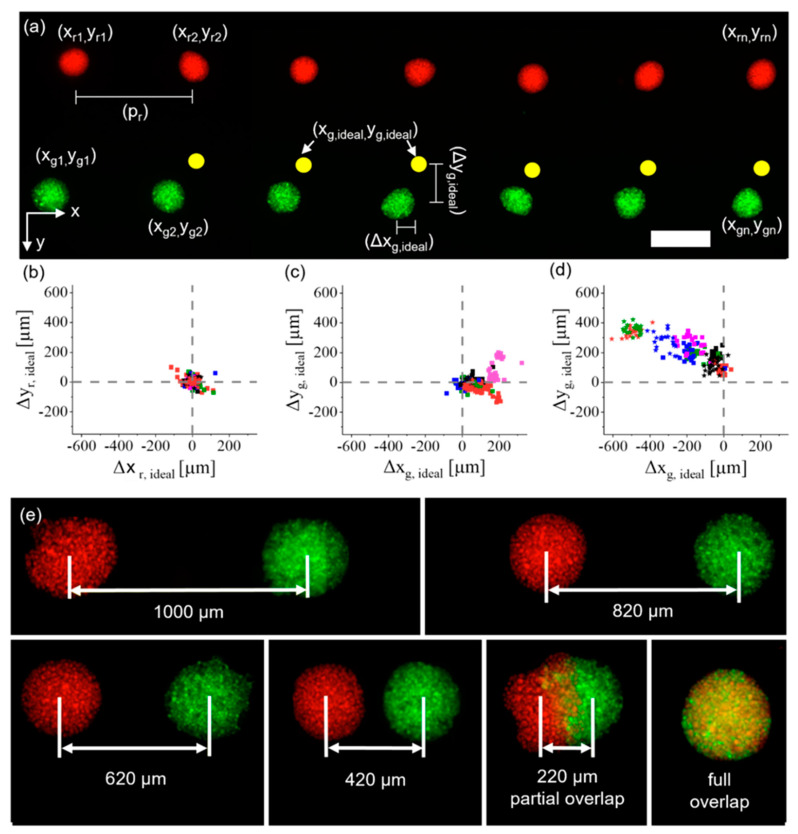
Characterization of the drop-on-demand (DoD) bioprinting process developed for multiple cell types. (**a**) Red- and green-labelled iMSCs, each color resembling one cell type, were printed with a pre-defined pattern and the deviation from the ideal position (yellow spots) was measured. Scale bar: 500 μm. (**b**) The deviations of red iMSCs were analyzed, and then (**c**) the deviations of green iMSCs before the samples were moved, and (**d**) samples were analyzed after they were removed and retransferred to the printing platform. Dotted lines show the ideal position. (**e**) iMSCs printed with decreasing pitches in approximately 200 μm steps up to an entire overlap.

**Figure 4 cells-12-00646-f004:**
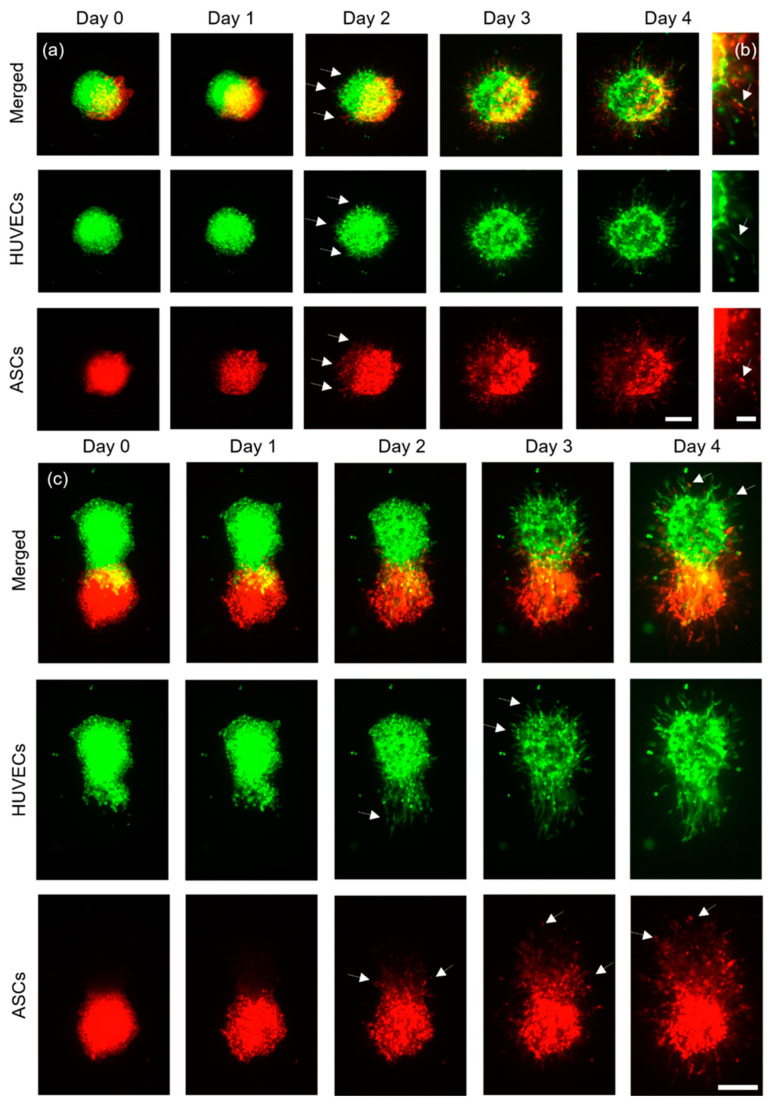
Representative images of ASCs stained with CellTracker^TM^ red, and HUVECs stained with CellTracker^TM^ green. (**a**) Aggregates printed with a full overlap (scale bar: 200 μm) were imaged for four days and (**b**) developed joint structures containing both cell types in the same structure (scale bar: 100 μm). (**c**) ASCs and HUVECs printed with a partial overlap are shown (scale bar: 200 μm).

**Figure 5 cells-12-00646-f005:**
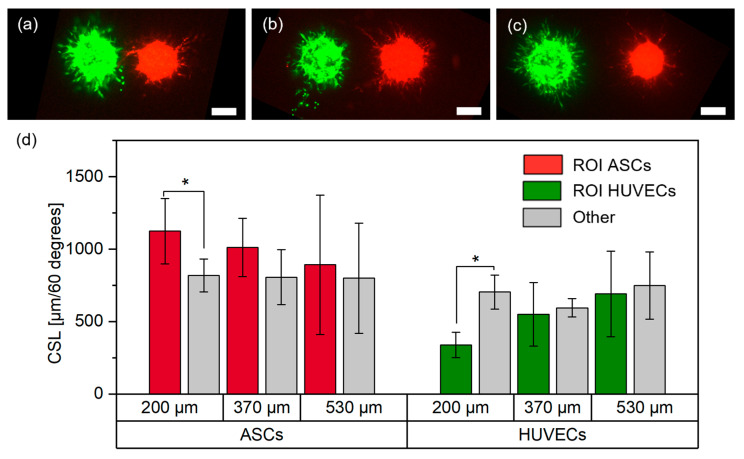
Representative images of ASCs stained with CellTracker^TM^ red, and HUVECs stained with CellTracker^TM^ green, in the (**a**) 200 μm, (**b**) 370 μm and (**c**) 530 μm group on day three (scale bar: 200 μm), and (**d**) respective measurements of the cumulative sprout(-like) lengths (CSLs) on day three; *, *p* < 0.05 using a paired Student’s *t*-test.

**Figure 6 cells-12-00646-f006:**
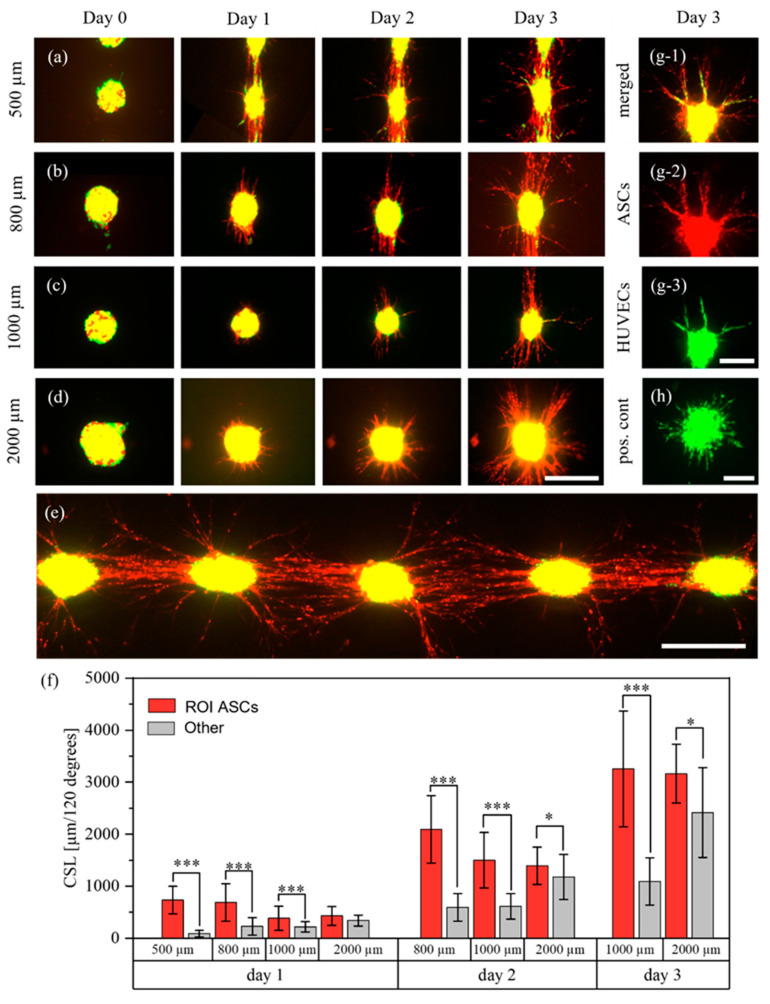
Behavior of ASCs stained with CellTracker^TM^ red, and HUVECs stained with CellTracker^TM^ green, in mixed aggregates. The cells were printed with different pitches of 500 μm (**a**), 800 μm (**b**), 1000 μm (**c**), 2000 μm (**d**) and imaged for three days during which the ASCs formed cellular bridges between the aggregates in the 500, 800 and 1000 μm groups. A representative image of several cell spots on day three, for a pitch of 800 μm, is shown in (**e**). The structures of the ASCs were measured every day and are displayed in (**f**); *, *p* < 0.05 and ***, *p* < 0.001 using a paired Student’s *t*-test. However, on day two in the 500 μm group and on day three in the 500 and 800 μm groups, measurements were no longer possible. This was due to too many cell structures which could not be separated visually. A representative image of an aggregate where HUVEC sprouts were aligned along the ASCs is shown in (**g**). (**h**) displays the behavior of HUVECs in the positive control. Scale bars a–e: 400 μm. Scale bars g,h: 200 μm.

**Figure 7 cells-12-00646-f007:**
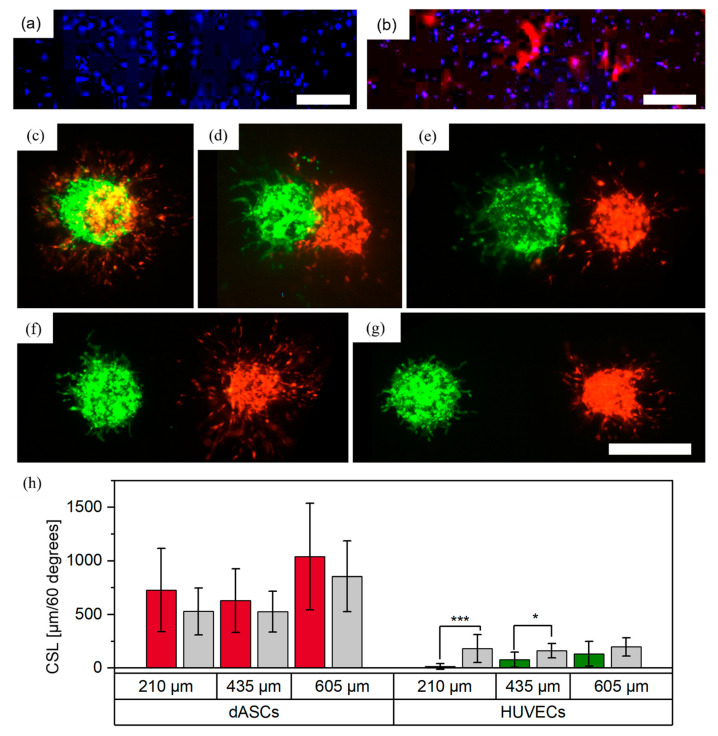
Printed patterns with dASCs stained with CellTracker^TM^ red, and HUVECs stained with CellTracker^TM^ green, as individual cell-laden droplets. ASCs were grown (**a**) without TGF-β1 and (**b**) with TGF-β1 and were stained with DAPI (blue) and with antibodies against αSMA (red). Scale bar: 500 μm. Aggregates were printed with (**c**) full overlap, (**d**) partial overlap, and distances of approximately (**e**) 210, (**f**) 435 and (**g**) 605 μm between the boundaries of the aggregates. Images were taken on day three. Scale bar: 500 μm. Measurements of the cumulative sprout(-like) lengths (CSLs) on day three are shown in (**h**); *, *p* < 0.05 and ***, *p* < 0.001 using a paired Student’s *t*-test.

**Figure 8 cells-12-00646-f008:**
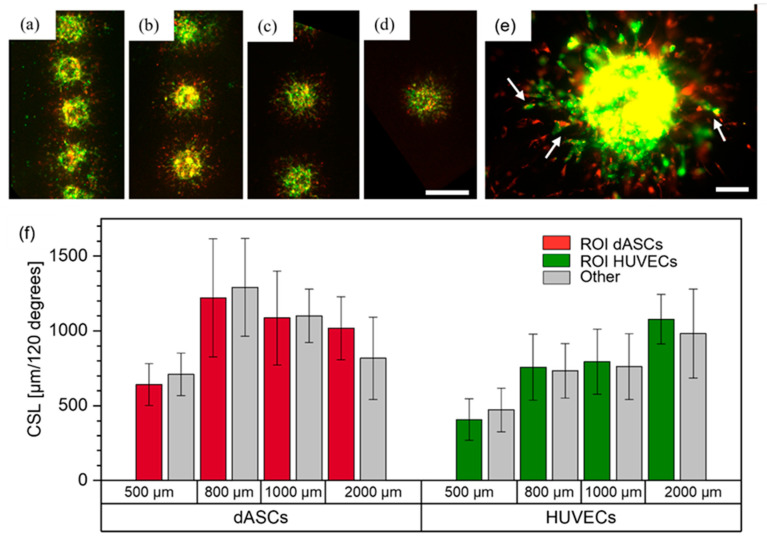
dASCs stained with CellTracker^TM^ red, and HUVECs stained with CellTracker^TM^ green, printed as mixed cell suspensions. Images from day three are shown for the (**a**) 500 μm, (**b**) 800 μm, (**c**) 1000 μm and (**d**) 2000 μm groups. Scale bar: 500 μm. (**e**) shows a mixed cell aggregate with joint sprout-like structures. Scale bar: 100 μm. (**f**) displays the measurements on day three of the cumulative sprout(-like) lengths (CSLs) for each cell type and each group.

## Data Availability

All data needed to evaluate the conclusions in the paper are presented in the paper and/or the Appendix A. Additional data related to this paper may be requested from the authors.

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
