# Peer review of "A Drop-on-Demand Bioprinting Approach to Spatially Arrange Multiple Cell Types and Monitor Their Cell-Cell Interactions towards Vascularization Based on Endothelial Cells and Mesenchymal Stem Cells"

_cells, 2023, doi:10.3390/cells12040646_

Round 1
Reviewer 1 Report
Line 52: There is a repeat: “as well as well”. Please correct and remove one of them.
Line 52: “In vivo” and “in vitro” should be written using Italics format. Also throughout the manuscript such as: line 47 and line 354.
Lane 132: Too many parentheses. Please reformulate to avoid this.
Lane 134: If “dASC” is the abbreviation for “ASCs differentiated to smooth muscle cells” then ”DASC” with capital “D” at the beginning of the sentence should be avoided.
Line 136: Please provide more details about the cultivation of iMSCs such as: origin, passages used etc.
Lane 172: “Results” should be corrected with “results”.
Lane 179 and lane 187: “imageJ” should be corrected with “ImageJ”.
Lane 209: Details about the microscope used in the experiments should be stated here also, similar with the ones in lane 234-235.
Lane 230 and 232: ‘4oC’ should be corrected using Superscript.
Lane 234: “H20” should be corrected using Subscript.
Lane 234: “ob-tained” should be corrected with “obtained”.
Lane 236: The chapter 3. “Results” should be changed in “Results and Discussion”. In my opinion, the authors have made plenty of discussion about their results and provided many comments regarding similar data from the literature in their field of interest, before the chapter 4. “Summary and Conclusions “.
Lane 317: The abbreviation “DoD” should be explained in the figure caption.
Lane 335: Please replace with the correct numbering: “2.3” should be replaced with “3.3”.
Lane 339: Please replace with the correct numbering: “2.3.1” should be replaced with “3.3.1”.
Lane 375: In the Figure caption, the color red and green should be explained. In my opinion, this would make it easier for the reader. The Figure should be comprehensive without the manuscript details.
Lane 392: Here as well, in the Figure caption, the color red and green should be explained. I am guessing red is for ASCs and green is for HUVECs, since HUVECs are the one that develop sprouts…
Lane 393: The abbreviation “CSL” should be explained in the figure caption.
Lane 463: In the Figure caption, the color red and green should be explained.
Lane 471: The first “Scalebars” should be replaced with “Scale bars”.
Lane 511: In the Figure caption, the color red and green should be explained.
Lane 516 and lane 539: The abbreviation “CSL” should be explained in the figure caption.
Lane 536: In the Figure caption, the color red and green should be explained.
Reviewer 2 Report
The authors proposed a drop-on-demand bioprinting approach, selected HUVECs and ASCs as model cells, printed HUVECs and ASCs via individual or mixed cell suspensions in various constellations, and found that the behavior of the mixed cell suspension (ASCs and HUVECs) was contrary to the behaviors of HUVECs and ASCs printed as separate droplets. Then the approach was applied to investigate the cellular behaviors of HUVECs and differentiated ASCs (dASCs) after printing. The experimental results showed that compared with the cell interaction between HUVECs and ASCs, each cell type had no inhibitory impact on the respective other cell type in mixed cell suspensions of HUVECs and dASCs. The proposed method did not require the labor and time-consuming process to fabricate spheroids, and could be used as a reference for studying the interactions between other cell types.
Overall, the article is well organized and its presentation is good. However, some issues still need to be improved:
(1) I suggest that the important influencing factors of this work should be discussed, such as the viscosity of bioinks, droplet shape and cell viability.
(2) In Section 3, it seems that the authors should give a comparative explanation for the difference in the behavior of the two mixed cell suspensions.
(3) Page 1, lines 28 & 29: "These findings demonstrate that our approach could be generally applied to investigate cell-cell interactions of different cell types in 3D co-cultures", the use of the word" generally " overstated the conclusions. For example, the study did not show the exact reasons for the cellular behavior that occurred after printing. Also, the reason for the anti directional formation of HUVEC sprouts was unclear.
(4) Page 3, line 108. The author mentioned that: "Based on the shape of the droplet", were there any satellite droplets produced during the experiment, and how were they eliminated?
(5) Page 6, line 256 & 257. " For this work, the cell types used are relevant for engineering vascularized tissue. However, the established method can be applied to various cell types and to more than two cell types " was mentioned. As we all know, the most important consideration for bioprinting is the type of cell involved in a particular cell. For example, endothelial cell model systems may require a different extracellular matrix material than muscle cells, while other cell types may require a support network of fibroblasts or stromal cells. In this study, the cell type used is associated with engineered vascularized tissue, there is a lack of explanation of other cell types.
(6) Compared to other bio-printing processes, drop-on-demand bioprinting approach requires much higher requirements for bioinks. In this experiment, how to determine the upper viscosity limit of bioinks? Furthermore, whether the effects of bioinks with different cell concentrations on droplet shape, velocity and printing stability were investigated.
(7) Page 9, line 335, the " 2.3. " and " 2.3.1." (page 10, line 339), the sequence labels are wrong.
(8) Page 12 &13, the behavior of the mixed cell suspension (ASCs and HUVECs) was analyzed, these observations were contrary to the behavior of HUVECs and ASCs printed as separate droplets, the authors concluded that it could be due to the interactions between cells. However, page 16 &17, about the conclusion on the behavior of mixed cell suspension (dASCs and HUVECS), the author thought that each cell type had no inhibitory effect on the respective other cell types, and their post-printing behavior was not necessarily dependent on adjacent cell aggregates. Why are these two conclusions so obviously different? The dASCs are formed by the differentiation of ASCs, for which the authors should give a comparative explanation.
(9) Page 18, line 570, "addresses several shortcomings of other methods to spatially align highly-dense cell aggregates" was mentioned, please list these disadvantages.
Round 2
Reviewer 2 Report
The revised manuscript is acceptable for publication in the journal.